# Development of Novel Thin Polycaprolactone (PCL)/Clay Nanocomposite Films with Antimicrobial Activity Promoted by the Study of Mechanical, Thermal, and Surface Properties

**DOI:** 10.3390/polym13183193

**Published:** 2021-09-21

**Authors:** Sylva Holešová, Karla Čech Barabaszová, Marianna Hundáková, Michaela Ščuková, Kamila Hrabovská, Kamil Joszko, Magdalena Antonowicz, Bożena Gzik-Zroska

**Affiliations:** 1Nanotechnology Centre, CEET, VŠB—Technical University of Ostrava, 17. Listopadu 2172/15, 708 00 Ostrava, Czech Republic; karla.cech.barabaszova@vsb.cz (K.Č.B); marianna.hundakova@vsb.cz (M.H.); michaela.scukova.st@vsb.cz (M.Š.); 2Faculty of Materials Science and Technology, VŠB—Technical University of Ostrava, 17. Listopadu 2172/15, 708 00 Ostrava, Czech Republic; 3Department of Physics, Faculty of Electrical Engineering and Computer Science, VŠB—Technical University of Ostrava, 17. Listopadu 2172/15, 708 00 Ostrava, Czech Republic; kamila.hrabovska@vsb.cz; 4Department of Biomechatronics, Faculty of Biomedical Engineering, Silesian University of Technology, Roosevelta 40, 41-800 Zabrze, Poland; Kamil.Joszko@polsl.pl; 5Department of Biomaterials and Medical Devices Engineering, Faculty of Biomedical Engineering, Silesian University of Technology, Roosevelta 40, 41-800 Zabrze, Poland; Magdalena.Antonowicz@polsl.pl (M.A.); Bozena.Gzik-Zroska@polsl.pl (B.G.-Z.)

**Keywords:** polycaprolactone, vermiculite, nanocomposites, thin films, antimicrobial activity

## Abstract

Infection with pathogenic microorganisms is of great concern in many areas, especially in healthcare, but also in food packaging and storage, or in water purification systems. Antimicrobial polymer nanocomposites have gained great popularity in these areas. Therefore, this study focused on new approaches to develop thin antimicrobial films based on biodegradable polycaprolactone (PCL) with clay mineral natural vermiculite as a carrier for antimicrobial compounds, where the active organic antimicrobial component is antifungal ciclopirox olamine (CPX). For possible synergistic effects, a sample in combination with the inorganic antimicrobial active ingredient zinc oxide was also prepared. The structures of all the prepared samples were studied by X-ray diffraction, FTIR analysis and, predominantly, by SEM. The very different structure properties of the prepared nanofillers had a fundamental influence on the final structural arrangement of thin PCL nanocomposite films as well as on their mechanical, thermal, and surface properties. As sample PCL/ZnOVER_CPX possessed the best results for antimicrobial activity against examined microbial strains, the synergic effect of CPX and ZnO combination on antimicrobial activity was proved, but on the other hand, its mechanical resistance was the lowest.

## 1. Introduction

Polycaprolactone (PCL) belongs to the aliphatic polyesters group, which have been the subject of increasing focus due to their biodegradability and biocompatibility [1]. Medical applications in particular, such as PCL applied in drug-delivery systems [2,3,4,5], medical devices using PCL for wound dressing [6,7,8], fixation devices [9] or in dentistry [10,11], as well as PCL included in tissue, bone, or blood vessel engineering [12,13,14,15,16,17], are the most investigated areas. PCL also plays an important role in the field of biodegradable and antimicrobial polymeric food packaging materials because of the increasing awareness of environmental problems with plastic waste [18,19,20,21].

PCL is a hydrophobic, semicrystalline polyester with a low melting point of about 60 °C and a glass transition temperature of about −60 °C, which allow easy processing. Moreover, its crystallinity tends to decrease with increasing molecular weight. The presence of repeating units of nonpolar methylene groups in its structure leads to properties typical for polyolefins, but on the other hand, the ester linkage causes its degradability. Its chemical structure is also behind its good miscibility with other polymers, with numerous applications in the biomedical field [1]. Nevertheless, PCL possesses some drawbacks, such as insufficient mechanical properties, lower barrier properties to gases, or low thermal stability, as well as a lack of bioactivity and low antimicrobial activity, which could be a limitation for many applications.

It is already known that the properties of polymers can be improved by the addition of nanofillers [22,23,24,25]. Some of the most progressively, and for the past few years extensively, studied nanofillers are clay minerals [26,27,28]—naturally abundant nontoxic layered silicates that are readily available at low prices, which makes them very attractive as nanofillers, and which can create three main types of composites with polymer chains: intercalated nanocomposites, exfoliated nanocomposites, or microcomposites. Compared to conventional composites, polymer/clay nanocomposites exhibit newly obtained and improved properties such as enhanced mechanical, thermal, barrier, electrical barrier, and optical properties; these structures also easily undergo degradation in the presence of microorganisms, due to the nanometric thickness of clay minerals, which are generally obtained with a lower silicate content below 5 wt%.

Besides these improvements, there is also a significant interest in the development of antimicrobial polymeric biomaterials for applications in the health or medical device, food, and personal hygiene industries [29]. For that reason, modification of polymeric matrices by antimicrobial species to prevent growth or reduce adhesion of microorganisms is a highly desired goal. When using nanocomposites, these interactions are more efficient because many biological structures also have properties corresponding to nanosized materials; therefore, clay minerals play the role of both nanofillers, improving the above-mentioned properties, and of carriers for antimicrobial agents. Since it is difficult to homogenously disperse natural clay minerals in an organic polymer matrix due to the hydrophilic nature of its surface, modifying clay minerals by cation exchange with organic compounds solves this problem to form a new suitable antimicrobial and well-dispersed nanofiller [29].

Although PCL itself and PCL nanocomposites are highly studied materials nowadays, there are very few reports in the literature dealing with antimicrobial PCL nanocomposites with clay-based nanofillers. Some frequently investigated organic antimicrobials intercalated into clay structures include quaternary ammonium salts. In their studies, authors Babu et al. [30,31], with the aim of developing antibacterial and antibiofilm surfaces, focused on the preparation of PCL/Cloisite 30B nanocomposites by solvent casting. The nanocomposite films prepared in this way showed antibacterial activity and acted as excellent barriers against the penetration of surrounding microorganisms. Thus, the desired properties were demonstrated even at a low concentration of filler, namely, 5 wt% and 1 wt%. Further work [32] dealt with biodegradable antimicrobial food packaging with the application of polymer nanocomposite film PCL/organoclay/chitosan. In this study, PCL samples were prepared with a nanocomposite organoclay filler, and chitosan was added as an effective antimicrobial component to enhance the antimicrobial properties of the biodegradable PCL composite film. It was found that while pure chitosan only showed antimicrobial effects against *E. coli*, the PCL nanocomposite film showed antimicrobial effects against *E. coli* and also against *P. aeruginosa* and *C. albicans*—but on the other hand, a sample with a higher chitosan content already showed low mechanical properties. The development of antimicrobial nanocomposite film PCL/clay mineral with increased mechanical ability and water vapor barrier properties for packaging applications has been addressed by Yahiaoui et al. [33]. As a result, it was found that when 3% organo-montmorillonite was added to the PCL matrix, growth inhibition was about 90% of that in the originally inoculated bacteria, increasing Young’s modulus as well as hardness, and decreasing water vapor permeability.

Microbial infections are becoming a growing problem and complication, especially in medicine; additionally, these pathogens are responsible for the contamination of food. Therefore, in this work, we focused on the preparation and characterization of thin PCL films with antimicrobial clay nanofillers, to extend the research into antimicrobial polymer/clay nanocomposites with the possibility of usage in the above-mentioned areas. This study aimed to obtain thin PCL nanocomposite films with good antimicrobial efficacy and enhanced mechanical and thermal properties. Based on our previous work in this area [34,35,36,37], we selected the clay mineral vermiculite as a carrier for antimicrobial components, which is able to effectively bind these compounds due to its high negative layer charge. As an active organic antimicrobial compound, the antifungal ciclopirox olamine (CPX) was used, which has been intercalated to vermiculite. For possible synergistic effects, a nanofiller in combination with the inorganic antimicrobial active ingredient zinc oxide (ZnO) was also prepared. Besides antimicrobial behavior, we also examined the structural arrangement of prepared samples and its influence on mechanical, thermal, and surface properties.

## 2. Materials and Methods

### 2.1. Materials

Clay mineral vermiculite (abbreviated VER) from a deposit in Brazil (Grena Co., Veselí nad Lužnicí, Czech Republic) was milled in a planetary mill, sieved, and the < 40 µm fraction used for experiments. Other chemicals used for clay nanofiller preparation were ciclopirox olamine (CPX) and ethanol as a solvent, and zinc oxide (ZnO) prepared from its precursor zinc chloride (ZnCl_2_) using sodium chloride (NaCl) and sodium carbonate (Na_2_CO_3_; all from Sigma Aldrich). Poly(ε-caprolactone) pellets (PCL; Sigma Aldrich, Prague, Czech Republic; Mw = 80,000 g/mol) and a chloroform (CHCl_3_; Mach Chemikálie Co., Ostrava, Czech Republic; purity 99.9%) solvent were used for preparation of thin PCL/clay nanocomposite films.

### 2.2. Preparation of Clay Nanofiller Containing Inorganic Antimicrobial Components

100 mL of 1 M sodium chloride solution were prepared by dissolving 5.85 g of NaCl in 100 mL of demineralized water, followed by heating to 80 °C. Furthermore, 5 g of VER, 2.5 g of ZnCl_2_ and 2.5 g of Na_2_CO_3_ were gradually added into this solution. After dissolution, an ultrasonic processor (UP100H from Hielscher, Teltow, Germany, 100 watts, 30 kHz) equipped with a Hielscher Sonotrode MS10 (titanium, for ultrasonic processor UP100H, for samples from approx. 20 mL up to 500 mL, Ø 10 mm, approx. 78.5 mm², approx. length 80 mm, amplitude approx. 70 µm (peak-through-value), male thread M6 x 0.75) was placed into the suspension; the whole mixture was sonicated for 15 min throughout the cycle at 50% amplitude.

After sonification, the sample was washed with demineralized water and centrifuged until the chlorides disappeared. The solid residue was dried in an oven at 80 °C and homogenized. The sample thus prepared was further subjected to calcination at 350 °C. The prepared nanofiller was named ZnOVER.

### 2.3. Intercalation of Organic Antimicrobial Component

The antimicrobial organic component CPX was intercalated into VER and ZnOVER powder samples by following procedure: VER or ZnOVER were suspended in 100 mL of demineralized water, then CPX at a 1:1 weight ratio with 50 mL of ethanol was added to these suspensions, and the mixtures were stirred and heated at 75 °C for 5 h. After intercalation, the mixtures were centrifugated and the solid residues were dried in an oven at 80 °C, followed by homogenization in a mortar into fine powders. The prepared nanofillers were named VER_CPX and ZnOVER_CPX.

### 2.4. Preparation of Thin PCL/Clay Nanocomposite Films

To prepare individual thin PCL/clay nanocomposite films, 1 g of PCL was weighed into the beaker and 20 mL of chloroform was added. The beaker was placed in an ultrasonic bath for 1 h. Then, 1 wt% of the respective nanofiller was added, and the beaker was again placed in the ultrasonic bath for 1 h. Due to the rapid evaporation of chloroform from the polymeric suspension, which heated spontaneously in the ultrasonic bath and subsequently concentrated, a small amount of chloroform was added to the suspension before pouring it onto the petri dish. Finally, each of the prepared suspensions of polymer nanocomposite in chloroform was equally poured into the thin layer of a petri dish and dried in an oven at 40 °C. The prepared thin PCL/clay nanocomposite films were named PCL/VER, PCL/ZnOVER, PCL/VER_CPX, and PCL/ZnOVER_CPX. Moreover, pure PCL thin film was prepared under the same conditions.

### 2.5. Sample Characterization

X-ray diffraction (XRD) patterns were measured using an X-ray diffractometer Rigaku Ultima IV (Rigaku, Japan, CuKα radiation, NiK_β_ filter, scintillation detector, Bragg–Brentano arrangement) in ambient atmosphere under constant conditions (40 kV, 40 mA, scanning range 1.5–40° 2θ, scanning speed 1.925°/min). CPX was dried at 75 °C, the same as other samples, before measurement and named CPXd. Commercial ZnO (nZ-BOCH01 from Bochemia a.s., CZ) was measured as ZnO control.

The IR spectra of powder clay nanofillers were measured by a potassium bromide pellets technique. Exactly 1.0 mg of sample was ground with 200 mg dried potassium bromide. This mixture was used to prepare the potassium bromide pellets. The IR spectra were collected using an FT-IR spectrometer Nicolet iS50 (ThermoScientific, Waltham, MA, USA) with DTGS detector. The measurement parameters were the following: spectral region 4000–400 cm^−1^, spectral resolution 4 cm^−1^; 64 scans; Happ–Genzel apodization. The IR spectra of the pure PCL thin film and thin PCL/clay nanocomposite films were measured by ATR (Attenuated Total Reflectance) techniques. The samples were laid and pressed by the pressure device on a single-reflection diamond ATR crystal. The IR spectra were collected using the FT-IR spectrometer Nicolet iS50 (ThermoScientific, USA) with DTGS detector on Smart Orbit ATR accessory. The measurement parameters were as follows: spectral region 4000–400 cm^–1^, spectral resolution 4 cm^−1^; 64 scans; Happ–Genzel apodization.

The specific surface area (SSA) of the powder clay nanofiller samples was measured under a nitrogen atmosphere using a Thermo Scientific Surfer (Thermo Scientific, Rome, Italy). The samples were degassed under vacuum (10^−6^ torr) at 80 °C for 24 h. The SSA was calculated using the BET (Brunauer–Emmett–Teller) equation. Particle size distribution measurements of the powder clay nanofiller samples were performed using a Horiba laser scattering particle size distribution LA-950 (Horiba, Japan) in distilled water. Particle size analysis was performed with refractive index values of 1.540 for VER, 1.500 for ZnO, 1.540 for CPX and 1.333 for distilled water. The ξ-potential measurements for powder nanofiller samples were performed using a Horiba Nanopartica SZ-100 analyzer (Horiba, Japan).

Surface morphology of all prepared samples was taken using a JEOL JSM-7610F Plus scanning transmission electron microscope (JEOL, France). The powder clay nanofillers were attached to the metal targets with carbon tape and were coated with a gold thin film to avoid electrical discharge during the observation. An accelerating voltage of 15 kV and a secondary electron detector (SEI) were used in the measurement. In the case of thin polymeric films, which were also attached to metal targets with carbon tape and covered by platinum powder, measurements were performed at an accelerating voltage of 15 kV and two types of secondary electron detectors were used (LEI: so-called lower detector, and SEI: so-called upper detector).

The water contact angle (WCA) of PCL materials was measured using a three-point technique at 23 °C, 993 mba, and a relative humidity of 45%. 0.1 mL of distilled water was deposited onto the surface of the PCL and PCL/clay nanocomposite thin films using a micropipette. Each drop (0.1 mL) was recorded using a Mitutoyo videocamera (Tokyo, Japonsko) and its images were evaluated using the Pixel Fox program (Germany). Contact angles were evaluated by the droplet tangent adjacent to the PCL surface. The examined WCAs are presented as a mean of 3 measurements per each PCL sample.

Thermal analysis of the pure PCL thin film and PCL/clay nanocomposite thin films was performed using a Setsys 24 Evolution Setaram thermal analyzer (Setaram, France). The thermal curves were recorded under following conditions: argon atmosphere (50 mL·min^−1^), final temperature of 1000 °C, heating rate of 10 °C min^−1^ and a sample mass of about 10 mg. Parameters such as final weight loss (∆ m), onset temperatures, and the temperature of maximum weight loss T_max_ were determined from thermal curves.

Differential Scanning Calorimetry (DSC) tests were performed in a DSC131 evo (Setaram, France) from 0 to 200 °C at a heating rate of 5 °C/min under argon atmosphere. The degree of crystallinity was calculated from the following equation, where ΔH_m_ is the specific melting enthalpy of the sample, w_PCL_ is the weight percentage of PCL and ${\Delta H}_{m}^{0}$ is the melting enthalpy of 100% crystalline polymer matrix (136.1 J/g for PCL [38]):(1)$${Χ}_{c}\left(\%\right)=\frac{{\Delta H}_{m}}{w_{\mathrm{PCL}}\times {\Delta H}_{m}^{0}}\times 100$$

The mechanical property analysis of the PCL thin film and PCL/clay nanocomposite thin films was carried out using the MTS Criterion Model 43 static testing machine. Each of the tested samples (with a 25 mm width, 13.2 mm thickness, and 500 mm length) were fixed in the grip in such a way that the measuring distance between the holders was the same for each sample. The device handle ensures accurate and reliable fixing of the axis of the test device without the possibility of displacement during the test. The PCL samples were subjected to a static tensile test at a speed of 50 mm/min. The results of force measurements were measured with an accuracy of 1N. The measurement of each sample was repeated 3 times.

### 2.6. Antimicrobial Tests

#### 2.6.1. Antimicrobial Tests of Clay Nanofiller Samples

The minimum inhibitory concentration (MIC) of the prepared powder clay nanofiller samples was determined by the lowest concentration of it that would completely inhibit microbe growth. Dilution and cultivation were performed on 96-well microtitration plates. The highest applied concentration was 10% (w/v) over water dispersion. This dispersion was further diluted by a threefold diluting method in glucose stock, in such a manner that the second to seventh set of hollows contained sample dispersed in concentrations of 3.33%, 1.11%, 0.37%, 0.12%, 0.041%, and 0.014%. The eighth set of wells contained pure glucose stock as a test control. A volume of 1 µL of glucose suspension of *Staphylococcus aureus* CCM 3953 (1.1 × 10^9^ cfu mL^−1^), *Escherichia coli* CCM 3954 (1.1 × 10^9^ cfu mL^−1^), and yeast *Candida albicans* ATC 90028 (1.1 × 10^9^ cfu mL^−1^), provided by the Czech collection of microorganisms (CCM), was applied into the hollows. A volume of 1 µL of yeast suspension was transferred (after 30, 60, 90, 120, 180, 240, 300 min and then in 24 h intervals for 5 days) from each well into 100 µL of fresh glucose stock and incubated in a thermostat at 37 °C for 24 and 48 h. Antimicrobial activity was evaluated by turbidity, which is a display of bacterial growth [39].

#### 2.6.2. Antimicrobial Tests of Thin PCL/Clay Nanocomposite Films

The pure PCL thin film and thin PCL/clay nanocomposite films were processed by a modified JIS Z 2801/ISO 22193 method (Measurement of Antibacterial Activity on Plastic Surfaces). The antimicrobial activity (AC) of all these samples was tested against the Gram-positive strain *Staphylococcus aureus* (*S. aureus*, CCM 3953), the Gram-negative strain *Escherichia coli* (*E. coli*, CCM 3954) and yeast Candia albicans (CCM 90028)—provided by the Czech collection of microorganisms (CCM).

Prior to testing, suspensions of tested strains were prepared in Erlenmeyer flasks and were incubated for 24 h in a thermostat at 35 ± 2 °C. To perform the test, the films were placed on sterile slides in petri dishes containing distilled water to maintain a higher humidity in the dishes. Suspensions of *S. aureus*, *E. coli*, and *Candida a.* strains at a density of 10^5^ CFU/mL were inoculated onto the surface of the tested films. After inoculation, the drops were covered with a polypropylene film and the plates were exposed to daylight. At the selected exposure time, the films were transferred to a labeled plastic container with a neutralizing solution in which they were shaken. A portion of this mixture was inoculated into a sterile petri dish and poured into 15 mL of TSA agar cooled to 45 ± 2 °C. Additional samples were taken at intervals of 24, 72, and 96 h. For better colony counting, a 10^−1^ dilution was made, and an aliquot taken from this dilution and embedded in TSA agar. All agars were cultured in a thermostat for 24 h at 35 ± 2 °C. After incubation, the number of grown colonies was evaluated for all agars.

## 3. Results and Discussion

### 3.1. X-ray Diffraction

The XRD pattern (Figure 1) of VER shows basal reflections at *2θ* 6.18° and 7.02° (*d*-values 1.428 nm and 1.258 nm), corresponding to the different hydration states of VER with two or one water layers around the hydrated interlayer cations [40]. The other VER reflections are at *2θ* 18.57°, 19.31°, 24.80°, 27.48°, 31.08°, 34.60°, 35.28°, and 37.63° (0.477 nm, 0.460 nm, 0.359 nm, 0.324 nm, 0.288 nm, 0.259 nm, 0.254 nm, and 0.239 nm) (ICSD PDF card no. 01-076-0847). Reflections at *2θ* 10.54°, 27.23°, 28.55°, 30.46°, 31.93°, and 33.12° (0.838 nm, 0.327 nm, 0.312 nm, 0.293 nm, 0.280 nm, and 0.270 nm) belong to the admixture phase of tremolite (ICSD PDF card no. 00-013-0437), which is evident for all samples. Reflection of quartz at *2θ* 26.69° (*d* = 0.334 nm) is also observed. The XRD pattern of ZnOVER shows a shift of VER basal reflections to *2θ* 7.28° and 7.81° (1.213 nm and 1.131 nm) as a result of ZnO formed in the interlayer space [41] and partial dehydration of VER structures after calcination at 350°C. ZnO nanoparticles anchored onto the VER particle’s surface are confirmed by reflections at *2θ* 31.82°, 34.27°, and 36.32° (0.281 nm, 0.261 nm, and 0.247 nm; ICSD PDF card no. 01-078-2585). VER_CPX shows new reflections at *2θ* 5.61°, 6.13°, 6.96°, 7.62°, and 9.58° (1.575 nm, 1.440 nm, 1.269, 1.160 nm, and 0.922 nm) confirmed irregularly arrangement of CPX intercalated into the VER interlayer space via intercalation and displacement of water molecules due to the occurrence of a dehydrated VER phase at *2θ* 8.90° (0.993 nm). Nevertheless, ZnOVER_CPX shows new reflections at *2θ* 6.24°, 6.67°, 6.91°, 7.59°, 9.54°, 11.63°, 12.32°, and 21.12° (1.416 nm, 1.325 nm, 1.279 nm, 1.164 nm, 0.925 nm, 0.761 nm, 0.718 nm, and 0.420 nm) and reflections of the dehydrated VER phase at *2θ* 8.81° and 17.98° (1.003 nm and 0.493 nm). It seems that CPX was intercalated more regularly, with better arrangement into ZnOVER in comparison with VER. The sharper form of ZnOVER_CPX peaks also reflects a more ordered structure. This can be influenced by the lower water content in the starting sample due to the calcination of ZnOVER at 350°C. Therefore, the lower content of interlayer water in sample ZnOVER positively influenced CPX intercalation. Moreover, reflections of non-intercalated CPX, mainly on the ZnOVER_CPX sample’s surface, are observed on the XRD patterns (marked *). However, these reflections show small shifts, and the disappearance of intensive reflections at 6.46°, 9.37°, 13.45°, 15.14°, and 15.44° is observed. These changes could be influenced by formation of CPX–Zn metal complexes [42], and CPX molecules can be also affected by dehydration conditions during sample preparation [43].

The XRD pattern (Figure 2) of pure PCL shows characteristic PCL reflections of the orthorhombic crystalline PCL structure at 21.38°, 22.05°, and 23.73° *2θ* (*d*-values 0.415 nm, 0.403 nm, and 0.375 nm), and a low, intense, wide reflection at 15.72° *2θ* (0.563 nm) confirmed a semicrystalline PCL structure [44,45]. XRD patterns of nanocomposite samples show other reflections with very low intensities due to the low content of fillers (1wt%). The PCL/VER sample shows reflections belonging to vermiculite at 6.19° (1.427 nm) and tremolite at 10.54° (0.839 nm). The XRD patterns of PCL/ZnOVER and PCL/VER_CPX show only subtle reflections for PCL/ZnOVER at 7.21° (1.224 nm) and PCL/VER_CPX at 6.77° (1.300 nm).

### 3.2. FTIR Spectroscopy

The FTIR spectrum of the initial VER (Figure 3), used as one of the nanofillers for PCL films, shows a band at 3673 cm^−1^ in the O-H stretching region attributed to structural OH groups, the band at 3411 cm^−1^ corresponding to an OH stretching vibration of adsorbed water and vibration at 1640 cm^−1^ belonged to the OH bending vibration of adsorbed water. Furthermore, the intensive band at 1003 cm^−1^ is assigned to Si–O stretching vibrations, together with an Si–O bending vibration at 447 cm^−1^ [46]. The FTIR spectrum of the antimicrobial organic component CPX (Figure 3,4) shows characteristic bands at 3136, 3057, 2925, 2853, and 2749 cm^−1^, corresponding to an O–H stretching vibration, aromatic C–H stretching vibrations, asymmetric and symmetric CH_2_ stretching vibrations, and a deformation overtone of CH_3_, respectively [47,48]. Absorptions at 1639, 1541, and 1511 cm^−1^ belong to Amide I and Amide II bands (C=O stretching and N–H deformation vibrations). The bands that occurred in the 1447–1354 cm^−1^ interval are due to the C–H deformation vibrations of CH_2_ and CH_3_ groups, and bands in the 1300–1130 cm^−1^ interval belong to the stretching vibrations of N–O and C–N bonds; there are also deformation vibrations of N–H that originate from primary aliphatic amine groups, and finally, further deformation vibrations of C–H [47,48].

The FTIR spectrum of ZnOVER nanofiller (Figure 4) also shows, in addition to the characteristic bands originating from VER, bands at 1480 and 1423 cm^−1^ corresponding to C=O vibrations belonging to traces of ZnCO_3_, which is an intermediate in the ZnO synthesis reaction.

In the spectrum of VER_CPX nanofiller (Figure 3), we can observe a shift in the region of characteristic valence vibrations of OH groups (3243 cm^−1^) from the initial VER and an enlargement in this region. These changes should confirm the binding of CPX, probably by hydrogen bonding between the structural OH groups of VER and OH groups of CPX. Similarly, this shift manifested in the characteristic stretching vibrations of OH groups in the ZnOVER_CPX sample (Figure 4, 3404 cm^−1^). Furthermore, we can observe for both types of nanofillers, that after CPX bonding there is a shift in the area corresponding to the stretching vibrations of the C=O group, which belongs to the amide arrangement directly connected to the shifted OH groups in CPX, also called an Amide II band. In the case of VER_CPX nanofiller, the shift is from the original value of 1541 cm^−1^ to 1555 cm^−1^ (Figure 3) and for the ZnOVER_CPX nanofiller (Figure 4) this value was shifted to 1557 cm^−1^. The presence of CPX in both types of nanofiller is also indicated by other adsorption bands characteristic for CPX, especially asymmetric and symmetric stretching vibrations of C–H bonds, deformation vibrations of C–H bonds, and stretching vibrations of N–O and C–N bonds. At the same time, when comparing the spectra of VER and ZnOVER, a decrease in the intensity of characteristic stretching and bending vibrations of OH groups corresponding to adsorbed water can be seen, which is due to the reduced water content in ZnOVER samples caused by calcination at 350 °C. In the same way, we can observe more intensive bands after CPX intercalation on ZnOVER compared to VER. These facts are in agreement with the results from X-ray diffraction analyses.

It is obvious, that from FTIR spectra of thin PCL/clay nanocomposite films (Figure 5a), we can only observe characteristic vibrations for PCL due to the predominant amount of PCL in all the thin PCL/clay nanocomposite films and only a small content of 1 wt% nanofiller, which was not reflected in the spectra; however, there is the possibility of comparing their intensities (Figure 5b). The main characteristic bands of PCL are 2944 and 2865 cm^−1^, belonging to asymmetric and symmetric stretching vibrations of CH_2_, 1720 cm^−1^ assigned to the stretching vibration of C=O, 1294 cm^−1^ corresponding to stretching vibrations of C–O and C–C in the crystalline phase, 1240 cm^−1^ assigned to asymmetric stretching vibrations of C–O–C bonds, and finally, 1160 cm^−1^ corresponding to stretching vibrations of C–O and C–C in the amorphous phase [49]. The more significant changes in intensities manifested themselves after the introduction of the given nanofiller only in the region of 1500–1000 cm^−1^ (Figure 5b), corresponding to the stretching vibrations of the C–O–C group. The most intense peaks belong to the spectrum of PCL/ZnOVER_CPX—this fact could be related to its arrangement.

### 3.3. Surface Characteristics and Morphology

One of the main surface properties used for clay nanofillers characterization was the specific surface area (SSA). SSA values for the initial VER (32.03 m^2^·g^−1^) and ZnOVER (29.11 m^2^·g^−1^) were almost the same (Table 1). A significant change in surface properties occurred after intercalation of the organic compound CPX, where a decrease in SSA values was evident—specifically for VER_CPX to 9.97 m^2^·g^−1^ and for ZnOVER_CPX to 9.16 m^2^·g^−1^ (Table 1).

The parameters of mode diameter (*d_m_*) and mean diameter (*d*_43_) were used to evaluate the particle size distribution analysis of clay nanofillers (Table 1). Due to the diverse nature of the particle size distribution (both modal and bimodal distributions) of the experimental samples, *d_m_* (mode) averages were used as core particle size values. In the case of bimodal distributions, two *d_m_* values are given (Table 1).

The initial VER nanofiller showed a monomodal particle size distribution with a mean diameter *d*_43_ = 4.64 μm. Ultrasonic growth of ZnO crystals led to the bimodal nature of the particle size distribution, with two maxima in mode diameters *d*_*m*1_ = 0.34 μm and *d*_*m*2_ = 6.72 μm for the ZnOVER sample. Intercalation of CPX into the VER structure led to an expansion of the particle size distribution of sample VER_CPX, which was reflected in an increase in *d*_43_ and *d_m_* values. The higher *d*_43_ value (12.8 μm) correlates with the structural behavior of VER particles, which was observed by SEM (Figure 6) and was already described in the study of [50], where the VER particles interacted with each other via their edges. A similar behavior was observed with the sample ZnOVER_CPX, where the size fraction *d*_*m*1_ = 0.34 μm was preserved, in contrast to the size of the majority fraction, which increased to *d*_*m*2_ = 11.52 μm. These distributional changes were also observed in SEM images (Figure 6), where the minor fraction surrounded the majority fraction.

The stability and prediction of particle agglomeration of powder clay nanofillers for the preparation of polymer films was evaluated using ζ-potential analysis. Table 1 shows that all prepared samples are very stable; their ζ-potential value is higher than −30 mV. The most stable particles formed the initial VER and intercalated VER_CPX. The presence of the ZnO nanoparticle inorganic phase led to a decrease in ζ-potential values. The ζ-potential values corresponded to the size changes of these samples. It is evident that with smaller particle sizes, the samples possess better surface stability.

Figure 6 shows SEM images of clay nanofillers, with each sample being displayed at a magnification of ×1500, ×5000, and ×10,000. In the case of the initial VER nanofiller we can observe irregularly shaped particles and individual VER plates. The presence of ZnO in the ZnOVER samples can be observed in the form of brighter, smaller agglomerating particles, especially at the edges of VER, which is in agreement with the bimodal character of particle size distribution (Table 1). SEM images of the VER_CPX sample show a more disturbed and delaminated structure than the initial VER, caused by intercalation of CPX. We can also see a smoother surface of individual plates with sharper edges. Similarly, in the case of ZnOVER_CPX, the addition of the organic phase led to a smoother surface of individual VER plates with very sharp contours, showing how larger particles are surrounded by smaller ones, which again agrees with particle size distribution.

Figure 7 shows SEM images of pure PCL thin film and thin PCL/clay nanocomposite films, where individual images are shown at magnifications of ×300 and ×1000, and images of water drops from water contact angle measurements.

From SEM images of the pure PCL thin film, we can observe a smooth structure formed by more-or-less regular hexagons of spherulites in a size range of 41–89 μm, which are characteristic for semicrystalline types of polymers and are formed during their crystallization [51,52]. Very fine particles are visible on the surface of the pure thin PCL film, such as sputtering residues, which did not affect the spherulite structure. In the case of PCL/VER thin nanocomposite film, the reduction in the size of individual spherulite grains to sizes in the range of 21–36 μm is evident, and they are rather more irregular than in pure PCL film. Furthermore, we can see larger cracks between individual spherulite formations, and a more detailed view (×1000) shows the presence of the clay nanofiller in the middle of the sperulitic grain. Similarly, irregular spherulitic grains in a smaller size range of 22–44 μm, compared to the pure PCL thin film, with larger voids between them are seen throughout the sample volume in the PCL/ZnOVER thin nanocomposite film. From SEM images (Figure 7) of the PCL/VER_CPX nanocomposite film, it is clear that the presence of another organic phase (CPX) caused the creation of a compact spherulitic structure with a smooth surface and no visible voids. In this case, we can observe only slight outlines of the original sperulitic structure of PCL with grains in a size range of 35–56 μm. Moreover, very fine VER_CPX nanofiller particles appear disordered on the surface of the thin PCL/VER_CPX film, both at the borders and in the centers of the PCL spherulite grains. Almost the same situation as in the PCL/ZnOVER film arose with PCL/ZnOVER_CPX, where the presence of ZnO underlined the growth of individual spherulitic grains from their centers, with large voids again appearing between the grains. The average grain sperulite size was in the interval of 15–51 μm. Overall, we can observe that nanofillers particles were very well incorporated into the PCL matrix, and caused the diminished structure of the PCL spherulitic grains; specifically, the ZnO nanoparticles caused irregularities in PCL grain shape and formation of larger voids between individual grains. The presence of an organic CPX phase smoothed out the film surface structure.

The surface stability (respectively wettability) of the pure PCL thin film and thin PCL/clay nanocomposite films was evaluated based on the water contact angles (WCA). Images of the water drops are shown in detail in Figure 7 and the measured data are in Table 2.

The pure PCL thin film has a hydrophilic surface with WCA 70.75°. The addition of nanofillers to the PCL matrix contributed to an increase in WCA values in the following order: 75.68° for the PCL/ZnOVER_CPX, <76.97° for the PCL/VER_CPX, <77.17° for the PCL/ZnOVER, and <80.78° for the PCL/VER. All thin PCL/clay nanocomposite films show hydrophilic surfaces. From the WCA values, it is evident that the presence of the organic phase (CPX) led to the preparation of a thin PCL/clay nanocomposite film with the highest surface wettability. The different spherulite grain sizes in PCL/VER_CPX and PCL/ZnOVER thin nanocomposite films could also have contributed to the lowest WCA values.

### 3.4. Thermal Analysis

Figure 8 shows TG and DTG curves of pure thin PCL film and thin PCL/clay nanocomposite films. Mass loss was determined in the temperature interval 230–520 °C (Figure 8a). The samples PCL, PCL/VER_CPX, and PCL/ZnOVER_CPX reached almost 100% mass loss. Slightly lower mass loss was observed with samples PCL/VER and PCL/ZnOVER (Table 3). As can be seen from DTG curves, mass loss in the above-mentioned temperature interval was realized at one step for samples PCL, PCL/VER, and PCL/ZnOVER, while in the case of samples with CPX it was two-step process due to the decomposition of CPX in the first step (Figure 8b). It is generally known that the introduction of inorganic material into polymeric matrices can improve their thermal stability, but in the case of ZnO the situation is completely opposite. We can observe decreasing thermal stability after incorporation of ZnO nanofillers into the original PCL, where T_max_ for PCL decomposition is 413.6 °C, 404.0 °C for PCL/ZnOVER, and slightly higher for PCL/ZnOVER_CPX due to presence of the organic compound CPX (407.3 °C; Figure 8b, Table 3). This behavior was described by Malakpour et al. [53], who said that ZnO can catalyze the decomposition of surrounding carbons in PCL.

Table 3 also summarizes the degree of crystallinity from DSC analysis of a pure polymer matrix of PCL and thin PCL nanocomposites. The most significant change occurred with nanofiller VER_CPX when compared to the pure PCL thin film: crystallinity of PCL/VER_CPX decreased from Χ_c_ = 63.0 to Χ_c_ = 53.0. Opposite to this, the highest crystallinity (Χ_c_ = 67.0) was possessed by sample PCL/ZnOVER_CPX. These results are in good agreement with the SEM analysis.

### 3.5. Mechanical Properties of Thin PCL/Clay Nanocomposite Films

The Young’s modulus (E), tensile strength (Rm), maximum force (Fmax) and maximum strain (Smax) parameter values describing the mechanical properties of pure thin PCL film and thin PCL/clay nanocomposite films were determined based on the static tensile test and are summarized in Table 4. As we can see, different values of mechanical properties were found for the pure thin PCL film and thin PCL/clay nanocomposite films. The clay nanofillers significantly affected both tensile strength values and Young’s modulus. They caused more than twice the decrease in values of the tensile strength of the thin PCL/clay nanocomposite films compared to the initial pure thin PCL film (23.9 MPa), where the lowest value was measured for the sample PCL/ZnOVER_CPX (10.7 MPa). A reduction in tensile strength to 14.8, 13.8, and 12.4 MPa, respectively, for PCL/VER, PCL/VER_CPX, and PCL/ZnOVER was also observed. The identical values as tensile strength (Rm) values were also found for maximum force (Fmax) and maximum strain (Smax).

From what is mentioned above, it can be assumed that the results of the measurements of mechanical properties based on the tensile test are probably influenced by the structural nature of thin PCL nanocomposite films, the size fraction of the nanofillers, and, mainly, by their ζ-potential values (Table 1). Individually, the influence of the organic phase (CPX), or on the other hand inorganic phase ZnO, on the tensile strength values has not been confirmed. Rather, we can see their synergistic effect in the case of sample PCL/ZnOVER_CPX, where the value of tensile strength was the lowest (10.7 MPa).

Differences were observed in the case of Young’s modulus (E), where it was found that the thin PCL/ZnOVER nanocomposite film was characterized by the highest stiffness (144 MPa). The lowest value of Young’s modulus was measured for the PCL/ZnOVER_CPX nanocomposite film (96 MPa).

It can be seen that the Young’s modulus (E) values correspond to changes in the crystallinity degree of the pure thin PCL film and thin PCL nanocomposites films, and at the same time can be influenced by the relatively identical size fraction of the ZnOVER (dm = 6.72 µm) and VER_CPX (dm = 7.18 µm) nanofillers.

### 3.6. Antimicrobial Tests

#### 3.6.1. Antimicrobial Test of Clay Nanofillers

Antimicrobial tests of clay nanofillers were performed against two bacterial strains *Staphylococcus aureus* and *Escherichia coli*, and yeast *Candida albicans*, and the activity at selected time intervals expressed by minimum inhibitory concentrations (MIC) is shown in Table 5.

At short time intervals, only the VER_CPX sample proved to be effective against all tested strains, with the best action against yeast *Candida a.*, mainly due to the antifungal nature of CPX. After 24 h and longer time intervals, the MIC values even decreased to a lowest concentration 0.014% w/v. Similarly, we found the lowest values of 0.014% w/v in the VER_CPX sample after 3 days against Gram-positive *S. aureus*, and for the Gram-negative *E. coli* strain (Table 5). Despite the antimicrobial behaviour of ZnO, ZnOVER samples showed modest activity against the investigated strains, and only in the longer time intervals, the best results were obtained against *E. coli* strain. The expected synergistic effect of ZnO and CPX antimicrobials in the ZnOVER_CPX sample was very poor. Since larger amounts of CPX were confirmed by organic carbon analysis, we also expected higher antimicrobial activity in the ZnOVER_CPX sample. On the other hand, this lower antimicrobial activity is probably associated with more intensive intercalation of CPX into the ZnOVER structure (confirmed by X-ray diffraction), and, consequently, poorer contact of very active CPX with bacteria.

#### 3.6.2. Antimicrobial Test of Thin PCL/Clay Nanocomposite Films

The antimicrobial activity of thin PCL/clay nanocomposite films was studied by a modified JIS Z 2801/ISO 22193 method (Measurement of Antibacterial Activity on Plastic Surfaces). Average numbers of colony forming units (CFU) at various time intervals are shown in Table 6.

Although the antimicrobial activity of the ZnOVER_CPX nanofiller was not so high (Table 6), the thin PCL film with this clay nanofiller—PCL/ZnOVER_CPX—exhibited the best activity against *E. coli*, even after 24 h, and against *Candida a.* after 72 h (Table 6). In the case of very resistant *S. aureus*, none of the examined samples failed to show antibacterial activity. The thin nanocomposite film PCL/VER_CPX also showed antimicrobial activity against *Candida a.*, but only after 96 h. The reason why PCL/ZnOVER_CPX has better antimicrobial activity than PCL/VER_CPX, despite the higher activity of clay nanofiller VER_CPX alone, could be the surface properties of this sample, where ZnO caused a less smooth surface with more irregularities, resulting in better contact with microbes. In addition, the low activity seen in all PCL/clay nanocomposite films can generally be caused by only a small amount of 1 wt% clay nanofiller. 

## 4. Conclusions

In this work, novel thin PCL nanocomposite films with antimicrobial nanofillers based on vermiculite, ciclopirox olamine and ZnO were prepared. The XRD and FTIR analyses confirmed the successful intercalation of antimicrobial compounds into the vermiculite structure, with a more regular arrangement of CPX in the case of the ZnOVER_CPX sample, due to the lower water content of the initial ZnOVER caused by its calcination. The values of specific surface area and particle size distribution parameters of prepared nanofillers showed that the presence of ZnO contributed to the bimodal character of particle size, that CPX caused decreases in SSA, and, finally, based on ζ-potential values we can conclude that all nanofillers were very stable. These findings were also supported by results from SEM image analysis.

In a very interesting way, the type of nanofiller affected the structure of thin PCL nanocomposite films, which is especially evident in the SEM images. The presence of nanofillers diminished the structure of spherulitic grains, the inorganic phase ZnO caused the irregular shape of spherulitic grains and larger cracks between the individual grains, in contrast to the organic CPX phase, which smoothed the structure of the films. All prepared PCL film had a hydrophilic surface.

Thermal analysis proved that after incorporating nanofillers based on ZnO into a PCL matrix, thermal stability of prepared thin PCL nanocomposite films slightly decreased compared to pure thin PCL film, because ZnO can catalyze the decomposition of surrounding carbons from PCL. The type of nanofiller used also significantly affected the mechanical properties of prepared thin PCL nanocomposite films, where we observed more than twofold decrease in values of the tensile strength in the thin PCL/clay nanocomposite films compared to the initial pure thin PCL film.

The antimicrobial activity of both powder nanofillers and the resulting thin PCL nanocomposite films tested against bacterial strains of *Staphylococcus aureus* and *Escherichia coli*, and against the yeast *Candida albicans*, revealed that the tested materials appear to be active mainly in longer time intervals. The thin film PCL/ZnOVER_CPX possessed the best antimicrobial activity—even better than PCL/VER_CPX, despite the higher activity of clay nanofiller VER_CPX alone, which could be caused by the surface properties of this sample, where ZnO caused a less smooth surface with more irregularities, resulting in better contact with microbes.

Overall, these novel thin PCL nanocomposite films represent a group of new promising materials with good prolonged antimicrobial behavior and have high potential for medical materials applications, as well as for the packaging industry.

## Figures and Tables

**Figure 1 polymers-13-03193-f001:**
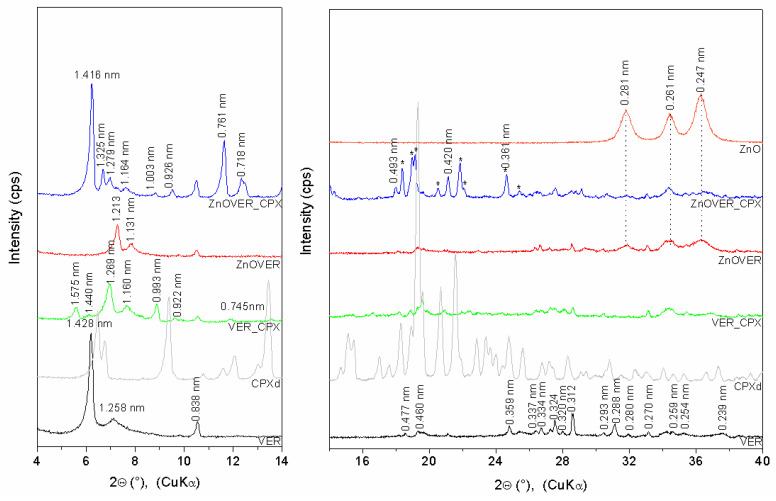
XRD patterns of initial VER and ZnOVER, VER_CPX, and ZnOVER_CPX nanofillers, and CPXd, ZnO.

**Figure 2 polymers-13-03193-f002:**
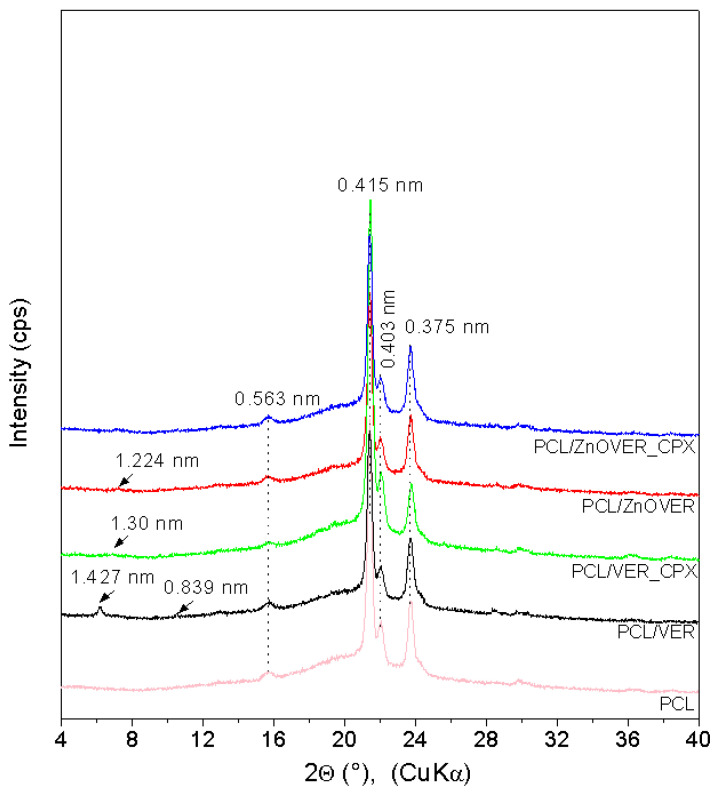
XRD patterns for pure PCL thin film and thin PCL/clay nanocomposite films.

**Figure 3 polymers-13-03193-f003:**
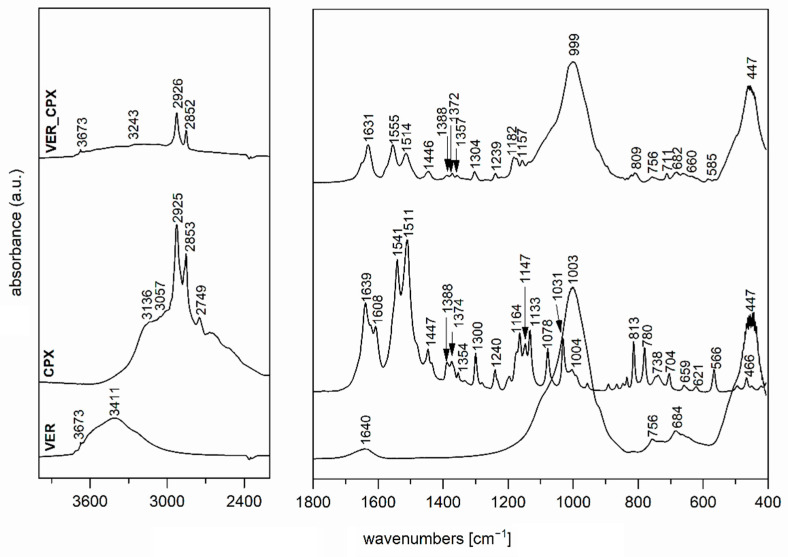
FTIR spectra of initial VER, CPX, and VER_CPX nanofiller.

**Figure 4 polymers-13-03193-f004:**
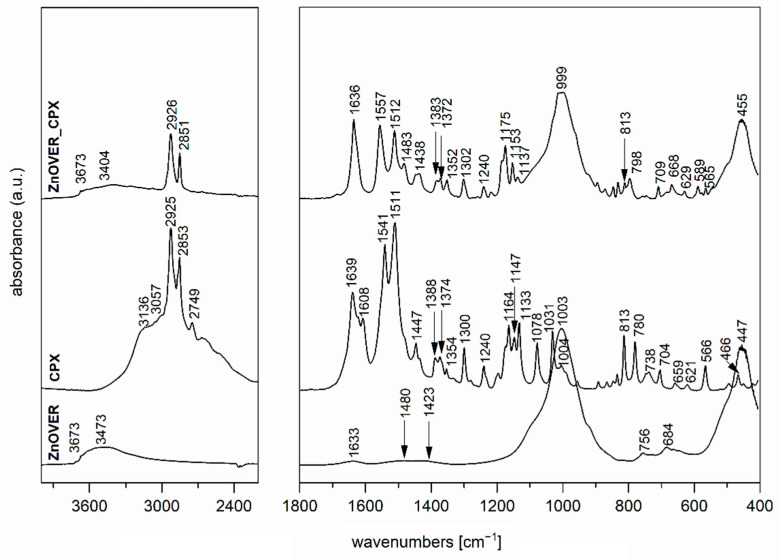
FTIR spectra of ZnOVER nanofiller, CPX, and ZnOVER_CPX nanofiller.

**Figure 5 polymers-13-03193-f005:**
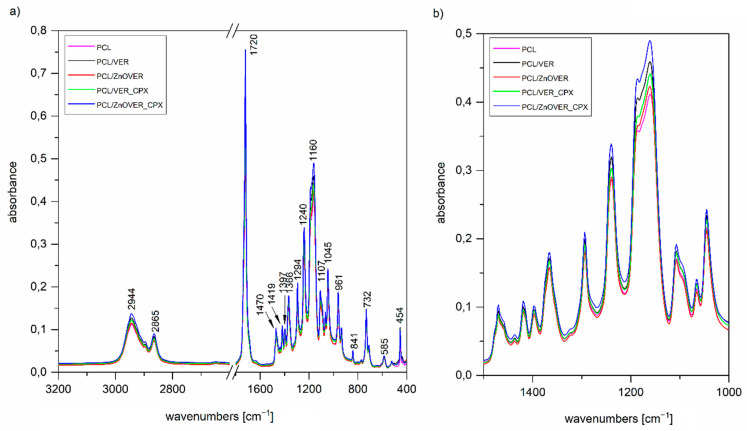
FTIR spectra of (**a**) pure thin PCL film and thin PCL/clay nanocomposite film in the region 3200–400 cm^−1^, (**b**) enlargement in the region 1500–1000 cm^−1^.

**Figure 6 polymers-13-03193-f006:**
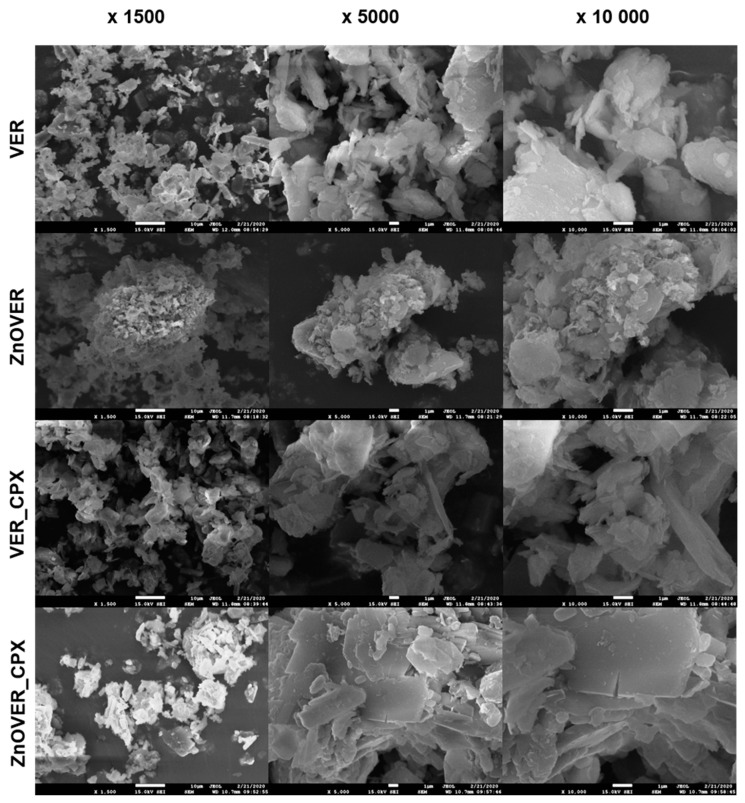
SEM images of VER, ZnOVER, VER_CPX, and ZnOVER_CPX nanofillers (magnification ×1500, ×5000 and ×10,000).

**Figure 7 polymers-13-03193-f007:**
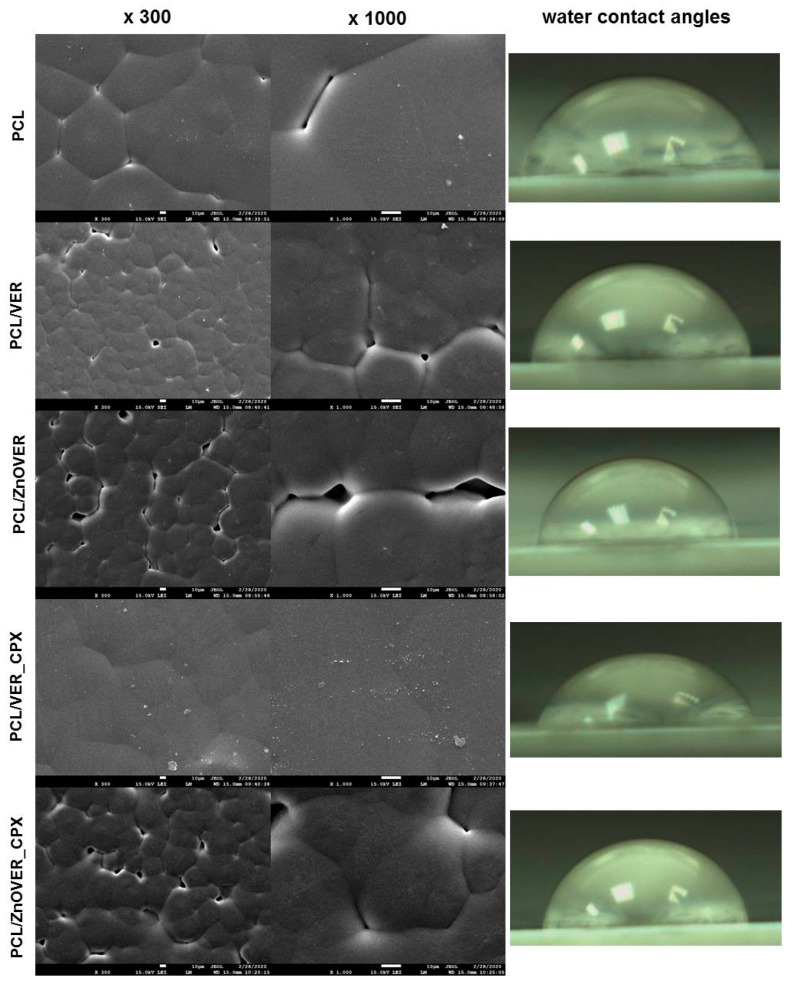
SEM images of pure PCL thin film and thin PCL/clay nanocomposite films (magnification ×300 and ×1000) with water contact angles.

**Figure 8 polymers-13-03193-f008:**
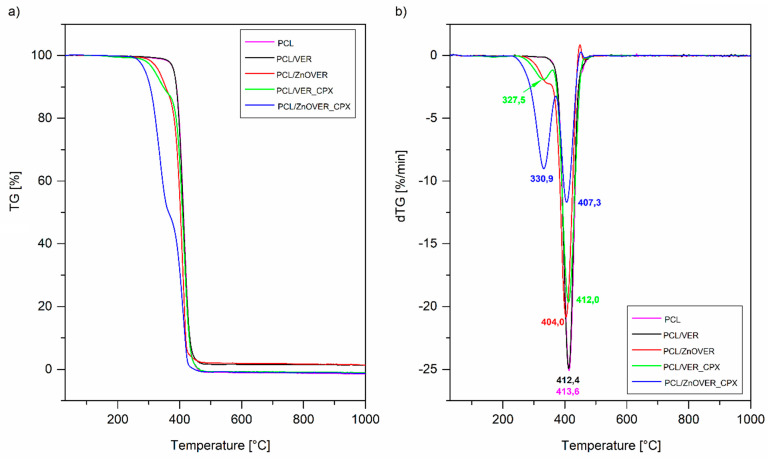
(**a**) TG and (**b**) DTG curves of pure PCL thin film and thin PCL/clay nanocomposite films.

**Table 1 polymers-13-03193-t001:** The specific surface area (SSA), particle size parameters (*d*_*m*_, *d*_43_), and ζ-potential values of clay nanofillers.

Sample	SSA [m^2^·g^−1^]	*d*_*m*_ [µm]	*d*_43_ [µm]	ζ-Potential [mV]
VER	32.03	4.75	4.64	−58.7
ZnOVER	29.11	0.34; 6.72	5.87	−34.8
VER_CPX	9.97	7.18	12.80	−44.1
ZnOVER_CPX	9.16	0.34; 11.52	13.25	−36.5

**Table 2 polymers-13-03193-t002:** Water contact angles (WCA) of the pure PCL thin film and thin PCL/clay nanocomposite films.

Sample	WCA [°]
PCL	70.75
PCL/VER	80.78
PCL/ZnOVER	77.17
PCL/VER_CPX	76.97
PCL/ZnOVER_CPX	75.65

**Table 3 polymers-13-03193-t003:** Thermal data of pure PCL thin film and thin PCL/clay nanocomposite films obtained from TG and DTG curves and crystallinity from DSC.

Sample	∆ m (%)	T_d_ (°C)	T_max_ (°C)	ΔH_m_ [J/g]	Χ_c_ [%]
PCL	99.9	380.0	413.6	85.8	63.0
PCL/VER	97.6	379.8	412.4	83.4	61.9
PCL/ZnOVER	97.3	371.8	404.0	79.9	59.3
PCL/VER_CPX	99.6	283.0, 376.0	327.5, 412.0	71.4	53.0
PCL/ZnOVER_CPX	100.0	286.5, 378.7	330.9, 407.3	90.3	67.0

**Table 4 polymers-13-03193-t004:** The results of the static tensile tests with standard deviations (S.D.) of the pure PCL thin film and thin PCL/clay nanocomposite films.

Sample	Young’s Modulus E (MPa)	Tensile Strength R_m_ (MPa)	Maximum Force Fmax (N)	Maximum Strain Smax (mm/mm)
PCL	129 ± 11	23.9 ± 5.0	41.0 ± 8.5	8.0 ± 0.7
PCL/VER	107 ± 8	14.8 ± 1.1	25.5 ± 2.1	5.3 ± 0.9
PCL/ZnOVER	144 ± 5	12.4 ± 2.2	21.5 ± 3.5	1.2 ± 0.6
PCL/VER_CPX	128 ± 5	13.8 ± 2.3	23.5 ± 3.5	3.6 ± 2.7
PCL/ZnOVER_CPX	96 ± 6	10.7 ± 1.3	18.5 ± 2.1	1.2 ± 0.3

**Table 5 polymers-13-03193-t005:** MIC values (%) (w/v) of clay nanofillers.

Strain	Sample	MIC [% w/v]					
		30 min	120 min	300 min	1 day	3 days	5 days
* **S. aureus** *	VER	-	-	-	-	-	-
	ZnOVER	-	-	-	-	1.11	1.11
	VER_CPX	10	10	10	1.11	0.014	0.014
	ZnOVER_CPX	-	-	-	0.37	0.37	0.37
* **E. coli** *	VER	-	-	-	-	-	-
	ZnOVER	-	-	-	1.11	1.11	1.11
	VER_CPX	3.33	3.33	3.33	3.33	0.014	0.014
	ZnOVER_CPX	-	-	-	3.33	1.11	1.11
* **Candida a.** *	VER	-	-	-	-	-	-
	ZnOVER	-	-	-	-	10	1.11
	VER_CPX	3.33	1.11	0.37	0.014	0.014	0.014
	ZnOVER_CPX	-	-	-	10	0.37	0.37

**Table 6 polymers-13-03193-t006:** Average numbers of fungal colony-forming units (CFU) of used strains at the various time intervals for thin PCL/clay nanocomposite films.

Strain	Sample	Number of CFU		
		Time of Contact [h]		
		24	72	96
** *S. aureus* **	PCL/VER	>330	>330	>330
	PCL/ZnOVER	197	>330	>330
	PCL/VER_CPX	165	>330	>330
	PCL/ZnOVER_CPX	26	>330	199
** *E. coli* **	PCL/VER	>330	>330	50
	PCL/ZnOVER	>330	>330	58
	PCL/VER_CPX	>330	>330	298
	PCL/ZnOVER_CPX	0	0	0
** *Candida a.* **	PCL/VER	71	>330	>330
	PCL/ZnOVER	93	>330	>330
	PCL/VER_CPX	135	74	0
	PCL/ZnOVER_CPX	22	0	0

## Data Availability

The data presented in this study are available on request from the corresponding author.

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
