# Peer review of "Development of Novel Thin Polycaprolactone (PCL)/Clay Nanocomposite Films with Antimicrobial Activity Promoted by the Study of Mechanical, Thermal, and Surface Properties"

_polymers, 2021, doi:10.3390/polym13183193_

Round 1
Reviewer 1 Report
Please see attached.
Reviewer 2 Report
The work entitled „Development of novel thin PCL/clay nanocomposite films with antimicrobial activity promoted by the study of mechanical, thermal and surface properties” raised very interesting topic concerning preparation and investigation polycaprolactone / clay composite materials. In the frame of work authors have used several measurement techniques, among others XRD, FT-IR spectrometry, laser scattering, scanning transmission electron microscopy and DSC. Authors achieved a good results in antibacterial films preparation. All the results obtained have been described very extensively and in details, also the English language is correct and comprehensible. Unfortunately, with all the undoubted advantages of this article, I have one major objection:
- The most important element of a scientific publication, which is the discussion of obtained results, is missing. The authors inserted sentences that could be considered as short, separate discussions, on each individual point in “Results” section. The article should be corrected in the way, that all its elements are related. Of course, the discussion can be joined with results (like it is in most of scientific articles), but the section must be clearly entitled "results and discussion", and discuss what arise from obtained results, of their comparison, etc.
- The entire article looks like a list of unrelated, separated sections. It must be corrected.
Round 2
Reviewer 2 Report
The chapter 3, should be entitled "Results and discussion",not "Results".
Author Response
Dear reviewer,
the chapter 3 was entitled as "Results and discussion".
Yours sincerely
Sylva Holešová